# Refining Endoscopic and Combined Surgical Strategies for Giant Pituitary Adenomas: A Tertiary-Center Evaluation of 49 Cases over the Past Year

**DOI:** 10.3390/cancers17071107

**Published:** 2025-03-26

**Authors:** Atakan Emengen, Eren Yilmaz, Aykut Gokbel, Ayse Uzuner, Sibel Balci, Sedef Tavukcu Ozkan, Anil Ergen, Melih Caklili, Burak Cabuk, Ihsan Anik, Savas Ceylan

**Affiliations:** 1Department of Neurosurgery, Bahcesehir University School of Medicine, 34734 Istanbul, Turkey; dratakanemengen@gmail.com; 2Department of Neurosurgery, VM Pendik Medical Park Hospital, 34899 Istanbul, Turkey; yilmazdreren@gmail.com (E.Y.); aykutgokbelnrs@gmail.com (A.G.); 3Department of Neurosurgery, Kocaeli University School of Medicine, 41380 Kocaeli, Turkey; ayse1994uzuner@gmail.com (A.U.); dr.melihcaklili@yahoo.com.tr (M.C.); cabukburak@yahoo.com (B.C.); drianik@yahoo.com (I.A.); 4Department of Biostatistics and Medical Informatics, Kocaeli University, 41380 Kocaeli, Turkey; b.s.balci@gmail.com; 5Department of Intensive Care Unit, VM Pendik Medical Park Hospital, 34899 Istanbul, Turkey; sedefto@gmail.com; 6Department of Neurosurgery, Kocaeli State Hospital, 41300 Kocaeli, Turkey; anilergen@gmail.com

**Keywords:** skull base, giant pituitary adenomas, endoscopic endonasal approach, pituitary, combined surgery

## Abstract

Giant pituitary adenomas (GPAs) are large tumors that extend into critical brain structures, making surgical removal challenging. While the endoscopic endonasal approach (EEA) is commonly used for pituitary tumors, achieving complete tumor removal in GPAs is often difficult. In some cases, additional transcranial approaches are required to improve resection while minimizing complications. In this study, we retrospectively analyzed 49 GPA cases treated in our center over the past year to evaluate the effectiveness of endoscopic and combined surgical approaches based on a classification system we previously defined. Gross total resection was achieved in 34.6% of patients, while near-total and subtotal resections were observed in 36.7% and 28.5%, respectively. While EEA was sufficient for many cases, some tumors required combined or transcranial approaches for better resection. Our findings emphasize the importance of a patient-specific, multimodal surgical strategy to optimize tumor removal and minimize surgical risks.

## 1. Introduction

Pituitary adenomas (PAs) are among the most common intracranial tumors, accounting for approximately 10−15% of all brain neoplasms [1,2]. These tumors arise from the anterior pituitary gland and exhibit a wide range of clinical presentations, including hormonal hypersecretion, pituitary insufficiency, and compressive symptoms due to mass effect. Most PAs are slow-growing and can be effectively managed with medical therapy, surgery, or radiotherapy depending on their size, functional status, and degree of invasiveness [3,4]. Giant PAs (GPAs), defined as tumors ≥ 4 cm in diameter, are relatively rare and account for approximately 5−14% of all PAs [5,6]. Due to their extensive parasellar and suprasellar extensions, GPAs present significant surgical challenges, often encasing critical neurovascular structures, such as the optic chiasm, cavernous sinus, and internal carotid arteries [7,8]. These tumors frequently present with visual impairment, pituitary dysfunction, and intracranial hypertension, necessitating an individualized surgical approach for optimal management [8,9]. The endoscopic endonasal approach (EEA) has become the preferred surgical technique for pituitary tumors, due to its minimally invasive nature, superior visualization, and reduced morbidity compared to traditional transcranial approaches [10,11]. However, achieving gross total resection (GTR) in GPAs remains challenging due to factors such as tumor consistency, cavernous sinus invasion, tumor volume and adherence to surrounding structures. When complete resection is not achievable with EEA alone, additional transcranial approaches may be required to maximize tumor removal while minimizing complications [12,13]. To overcome these challenges, classification systems have been developed to assess resection feasibility and guide surgical decision-making [14,15,16]. Recently, we introduced a landmark-based classification system to categorize GPAs according to their anatomical extension and surgical accessibility. This system offers a structured framework for selecting the most effective surgical approach [17].

In this study, we retrospectively analyzed 49 patients with GPA treated at our tertiary center over the past year to assess the impact of endoscopic and combined surgical approaches on resection outcomes. Using our previously defined classification system, we aimed to identify the most effective strategies for maximizing tumor removal while minimizing morbidity. Our findings contribute to the growing body of evidence underscoring the importance of a patient-specific, multimodal approach to the surgical management of GPAs [17,18,19,20].

## 2. Materials and Methods

Among the 517 histopathologically confirmed PAs treated by a single surgeon at the Bahcesehir University and Kocaeli University Department of Neurosurgery between September 2023 and September 2024, we included 49 patients whose tumors had a maximum diameter exceeding 40 mm in a single axis on preoperative MRI. However, two patients who did not attend follow-up visits were excluded. The demographic data, clinical symptoms, imaging findings, surgical details, and follow-up outcomes of the remaining 49 patients were retrospectively analyzed. Postoperative follow-ups were conducted until February 2025, with a minimum follow-up period of four months. Follow-ups included endocrinological and clinical evaluations, along with MRI assessments conducted at four months postoperatively.

### 2.1. Clinical, Endocrinological, and Histopathological Evaluation

Headache was excluded from the study as it is a nonspecific symptom. Visual function was evaluated preoperatively and postoperatively by an independent ophthalmologist using visual-field testing, with findings categorized as right, left, or bilateral involvement. These assessments were crucial in evaluating the impact of tumor size and surgical intervention on visual function recovery.

All tumor specimens underwent histopathological and immunohistochemical analysis to confirm the diagnosis of pituitary adenoma and assess multiple pituitary hormone stains, apoplexy, proliferative activity, P53 expression, and the Ki-67 index (≤3% or >3%). The assessment of these markers provided insights into tumor aggressiveness and potential recurrence risk, contributing to a more comprehensive understanding of tumor behavior.

Hormonal evaluations, including preoperative and postoperative basal and functional pituitary hormone assessments, were conducted by an independent endocrinologist. For secreting adenomas, remission at the third postoperative month was determined using established consensus criteria from the literature [21,22,23,24]. These criteria incorporated biochemical markers, hormone suppression tests, and clinical symptom resolution to accurately determine the remission status.

### 2.2. Radiological Evaluation

Preoperative, postoperative, and follow-up MRI scans were independently evaluated by a neuroradiologist. Preoperative MRI was used to classify tumors based on size (40–50 mm and >50 mm) and shape (dumbbell-shaped, multilobular, or round). Tumor volume was primarily assessed based on its predominant location and extension, categorized into the chiasmatic cistern, sphenoid sinus, lateral ventricle inferior wall, anterior cranial fossa, retroclival region, and temporal lobe. Cavernous sinus invasion was assessed using the Knosp classification and categorized as no invasion, Knosp grade 1/2, and Knosp grade 3/4 [25,26]. Additionally, the suprasellar extension was assessed based on tumor height exceeding 2 cm and the suprasellar-to-infrasellar volume ratio, categorized as greater or less than one.

Tumor volume was categorized into Zone 1, Zone 2, and Zone 3 based on the landmark-based classification previously defined in our study [17] (Figure 1). This classification system enabled a standardized assessment of the tumor location and its spatial relationship with critical anatomical structures, providing a more objective evaluation of surgical strategy and resection outcomes. Complex cases were defined as tumors with Knosp grade 3/4 invasion or those located in Zone 3. The first postoperative MRI was performed within 24 h of surgery, with a follow-up MRI assessment conducted three months later. The extent of resection was determined based on the MRI performed within 24 h of surgery.GTR: Defined as the absence of residual tumor on postoperative MRI.Near-total resection (NTR): Defined as a ≥90% reduction in tumor volume.Subtotal resection (STR): Defined as a <90% reduction in tumor volume.

#### 2.2.1. Surgical Technique

All procedures were conducted using an image-guided neuronavigation system. The standard EEA was the most commonly used technique in our series. However, alternative endoscopic approaches were utilized based on tumor extension. Specifically, an extended EEA was performed for tumors extending into the anterior cranial fossa, while a transpterygoid approach was employed for cases with Knosp grade 3/4 cavernous sinus invasion. The details of these surgical approaches have been previously described [17,27,28,29,30].

In the initial stage, after opening the sellar dura, the inferior portion of the tumor and any cavernous sinus components, if present, were decompressed. Early evacuation of the inferior portion prevented premature descent of the suprasellar cistern, facilitating complete resection of the posterior suprasellar component. After internal tumor debulking, anatomical structures were carefully delineated, and arachnoid and capsular dissection were performed when feasible. The suprasellar portion was resected in a posterosuperior-to-anterior direction, preserving a thin capsule to facilitate internal decompression and NTR. These techniques were primarily applied to the Zone 1 and Zone 2 tumors.

For Zone 3 and complex tumors, an additional transcranial approach was employed in selected cases. Tumors extending into the anterior skull base, temporal lobe, or lateral ventricle were managed using simultaneous or staged frontotemporal, transcallosal, or pterional craniotomy combined with endoscopic surgery (Figure 2). Following the dural opening, meticulous microsurgical dissection was performed to separate the tumor from the optic apparatus, major vascular structures, and surrounding brain parenchyma. Internal debulking was performed using ultrasonic aspiration or suction, facilitating safer capsular dissection.

Preservation of the pituitary stalk and hypothalamic structures was prioritized whenever feasible. Meticulous hemostasis was achieved, and dural reconstruction was performed using a galeal graft to minimize the risk of cerebrospinal fluid (CSF) leakage. For sellar reconstruction, a combination of a fascia lata graft, adipose tissue, and a nasoseptal flap was utilized to ensure optimal closure and skull base support. In selected cases, lumbar drainage was administered and maintained for 96 h postoperatively when deemed necessary to reduce the risk of CSF leakage and facilitate healing.

#### 2.2.2. Statistical Analysis

All statistical analyses were performed using IBM SPSS 29.0 (IBM Corp., Armonk, NY, USA). The Shapiro–Wilk test was used to assess the normality assumption. Continuous variables were presented with mean ± standard deviation or median with interquartile range (IQR). Categorical variables were summarized as counts and percentages. Group comparisons for continuous variables were performed using the Kruskal–Wallis test. Associations between categorical variables were analyzed using the chi-square test. A *p*-value < 0.05 was considered statistically significant.

## 3. Results

A total of 49 patients with GPA were included in the study and classified into three groups based on the extent of resection: STR (28.6%), NTR (36.7%), and GTR (34.7). The mean age of the cohort was 45.57 ± 12.89 years. The distribution of age groups was as follows: 36.7% were under 40 years, 49% were between 40 and 60 years, and 14.3% were above 60 years. There was no significant difference in gender distribution (*p* = 0.562), with males comprising 71.4% of the overall cohort.

### 3.1. Preoperative Clinical and Endocrinological Features

Bitemporal hemianopsia was the most frequent visual impairment (69.4%), while unilateral right (20.4%) and left (10.2%) visual loss were less common. The distribution of visual impairment was similar across the three groups. Recurrent adenomas were observed in 14.3% of the cases, with a relatively higher proportion in the STR group (28.6%) compared to the NTR (11.1%) and GTR (5.9%) groups. Regarding adenoma functionality, 65.3% of the tumors were non-secreting, while 34.7% were secreting. Among the secreting adenomas, 26.5% were prolactin-secreting (the most common), 4.1% were GH-secreting, 2% were ACTH-secreting, and 2% were TSH-secreting. Preoperatively, 49% of patients had some degree of hormonal deficiency, with the highest incidence observed in the GTR group (64.7%). Among these patients, 26.5% exhibited panhypopituitarism, 18.4% hypogonadism, and 4.1% hypocortisolism.

### 3.2. Radiological Tumor Characteristics and Surgical Outcomes

Tumor characteristics were analyzed based on size, shape, and extension. The mean maximum tumor diameter was 47.8 ± 6.3 mm, with a median of 46 mm (range: 40–65 mm). Tumors were further classified by shape: 51% were multilobular, 36.7% were round, and 12.2% were dumbbell-shaped. Regarding cavernous sinus invasion, 63.3% of tumors were classified as Knosp grade 3–4, 32.7% as Knosp 1–2, and 4.1% exhibited no invasion. Among two patients with Knosp 0 adenomas, one underwent GTR, while the other underwent STR. The patient who underwent STR had an isolated suprasellar multilobulated adenoma. The suprasellar extension was observed in 61.2% of tumors measuring <2 cm, while 38.8% exceeded this threshold. Tumor localization was assessed, revealing extension into the chiasmatic cistern in 34.7% of cases, the sphenoid sinus in 16.3%, and the lateral ventricle in 18.4%. Smaller proportions extended into the anterior cranial fossa (12.2%), retroclival region (10.2%), and temporal lobe (8.2%). For Zone 1 tumors, GTR is 100% (8/8). In Zone 2, GTR is 28.6% (6/21), NTR 47.6% (10/21), and STR 23.8% (5/21). In Zone 3, GTR drops to 15% (3/20), while NTR and STR rise to 40% (8/20) and 45% (9/20), respectively. These findings indicate a decline in complete resection from Zone 1 to Zone 3, with incomplete resection rates increasing as tumor invasiveness and extension progress (71.4% in Zone 2, 85% in Zone 3).

The endoscopic endonasal transsphenoidal (EEA) approach was the most commonly employed surgical technique, used in 83.7% of cases. A combined endoscopic and transcranial approach was necessary in 16.3% of the cases, primarily for Zone 3 tumors and those with complex extensions. Transcranial approaches, such as frontotemporal, transccallosal or pterional craniotomy, were performed in selected cases involving significant lateral or anterior cranial fossa invasion. Notably, none of the patients in the GTR group required a combined approach, whereas 28.6% of STR cases and 22.2% of NTR cases did. Intensive care unit admission was necessary for 20.4% of patients.

### 3.3. Parameters Affecting Resection

Resection rates were higher for Zone 1 and Zone 2 tumors, whereas the Zone 3 tumors and those with Knosp grade 3–4 invasion had a lower likelihood of complete resection. Tumor complexity had a significant impact on the extent of resection (*p* < 0.001). Simple tumors accounted for 82.4% of GTR cases, whereas complex tumors were predominant in the STR (92.9%) and NTR (72.2%) groups. The tumor shape also played a crucial role in the resection outcomes. Multilobulated tumors were more common in the STR (78.6%) and NTR (61.1%) groups, while round tumors were primarily observed in the GTR group (70.6%) (*p* = 0.003). Suprasellar extension significantly influenced resection rates. Large suprasellar tumors (>2 cm) were found in 71.4% of STR cases but only in 5.9% of GTR cases (*p* = 0.001). Similarly, zone-based tumor classification revealed that Zone 1 tumors were exclusively found in the GTR group (47.1%), while Zone 3 tumors were predominant in the STR group (64.3%) (*p* < 0.001). Knosp grade 3−4 invasion was observed in 63.3% of cases, with the highest prevalence in the STR group (92.9%), followed by the NTR (61.1%) and GTR (41.2%) groups. Similarly, cavernous sinus involvement showed a similar pattern, with more extensive invasion observed in cases of incomplete resection (Table 1).

### 3.4. Histopathological Characteristics and Postoperative Outcomes

Based on the Ki-67 proliferation index, 71.4% of tumors had a Ki-67 ≤ 3%, while 28.6% exhibited Ki-67 > 3%. P53 positivity was observed in 36.7% of cases, with a comparable distribution among STR (35.7%), NTR (44.4%), and GTR (29.4%) groups (*p* = 0.702). Increased tumor proliferation activity was observed in 44.9% of patients, with the highest rate in the NTR group (55.6%); however, the difference was not statistically significant (*p* = 0.300). Apoplexy was observed in 20.4% of patients, with the highest incidence in the GTR group (29.4%); however, the distribution did not differ significantly among the groups.

Secreting adenomas were assessed based on the remission criteria in the third postoperative month. Among the 17 patients with functioning adenomas, 23.5% achieved remission, while 76.5% exhibited persistent hormone elevation. New-onset postoperative endocrine deficiency occurred in 10.2% of cases, including 2% panhypopituitarism and 8.2% hypocortisolism. It was slightly more frequent in the STR (21.4%) group compared to NTR (5.6%) and GTR (5.9%) groups, though the difference was not statistically significant. Complications included transient diabetes insipidus (9/49), cerebrospinal fluid leakage (2/49), apoplexy (2/49), hypocortisolism (3/49), epidural hematoma (1/49), and epistaxis (1/49). There were no perioperative mortalities.

## 4. Discussion

GPAs pose significant surgical challenges due to their extensive suprasellar and parasellar extension, potential cavernous sinus invasion, and involvement of critical neurovascular structures. The optimal surgical approach for these tumors is still debated, particularly in terms of the feasibility of GTR and the need for combined techniques. Our retrospective analysis of 49 cases with GPA offers valuable insights into the surgical outcomes and prognostic factors affecting the extent of resection.

PA typically presents with headaches, visual impairment, or endocrinopathy [31,32,33]. Although generally benign, some cases exhibit more aggressive behavior, characterized by bone destruction and rapid growth patterns [34,35]. In such cases, compression of critical anatomical structures can result in life-threatening complications. The primary goal of surgery is to maximize tumor resection while ensuring adequate optic nerve decompression. In recent years, EEAs have become increasingly popular for their wide panoramic view, improved anatomical delineation, and lower morbidity and mortality rates [17,19,28,36,37]. However, transcranial surgery remains essential for cases involving cavernous sinus invasion or extensive suprasellar extension. A comprehensive preoperative radiological evaluation is crucial for determining the most appropriate surgical approach, enabling a more tailored approach to tumor resection and patient management [38,39]

A thorough preoperative analysis of pituitary MRI scans is essential for surgical planning. Key factors, including maximum tumor diameter, morphology, suprasellar extension and height, and cavernous sinus invasion, must be carefully assessed for their impact on resection outcomes. Therefore, several classification systems have been proposed to improve giant adenoma (GA) stratification and guide surgical planning [14,15,16]. In our latest study, we introduced two classification systems—the morphological score and the landmark-based classification—to enhance tumor stratification and guide surgical decision-making [17]. Our findings indicate that GTR was achieved in 34.7% of cases, while NTR and subtotal resection (STR) were performed in 36.7% and 28.6% of cases, respectively. The likelihood of achieving GTR was significantly impacted by tumor morphology, Knosp grade, and suprasellar extension. Specifically, round-shaped tumors had a higher probability of complete resection compared to multilobulated or dumbbell-shaped tumors (*p* = 0.003). This finding aligns with previous reports indicating that irregular tumor borders and multiple lobulations impede total removal. Similarly, a suprasellar extension > 2 cm was associated with significantly lower GTR rates, likely due to adhesions to the optic chiasm and hypothalamic structures (*p* = 0.001).

Several studies have highlighted cavernous sinus invasion as a key factor influencing resection outcomes, often necessitating adjuvant radiotherapy or a transcranial approach for optimal management [25,29,40,41]. Tumors invading the lateral compartment of the cavernous sinus or extending superolaterally have lower resection rates, emphasizing the surgical challenges associated with such invasions [26,42]. Cavernous sinus invasion is a key factor influencing surgical outcomes. In our series, 63.3% of the tumors exhibited Knosp grade 3–4 invasion, with higher Knosp grades correlating with lower resection rates. Although previous studies indicate that Knosp grading alone does not preclude GTR, our findings underscore the considerable challenge of achieving complete resection in cases with extensive cavernous sinus involvement. With increasing surgical experience, the resection rates of tumors extending into the temporal lobe or retroclival region via the EEA have improved (Figure 3). However, for tumors with suprasellar extension into other anatomical regions, additional transcranial surgery may still be required.

In our study, EEA alone was effective for tumor removal in 83.7% of the cases. However, in 16.3% of patients—particularly those with Zone 3 tumors, anterior cranial fossa invasion, or significant lateral extension—a combined endoscopic and transcranial approach was necessary. These findings align with the growing consensus that a multistaged or combined approach may be essential for complex GPAs to optimize resection while minimizing neurological complications. EEA alone may be ineffective in cases where the tumor exhibits irregular multilobular morphology, a narrow diaphragmatic opening, fibrous consistency, lateral extension beyond the optic apparatus or cavernous sinus, or asymmetric neocortical lobe involvement, all of which hinder tumor descent and resection. In such cases, combining a transcranial approach with EEA can help overcome surgical challenges, prevent tumor apoplexy and intratumoral hemorrhage, and ultimately optimize surgical management and patient outcomes [6,18,43,44,45,46]. A single-stage combined approach is recommended due to its shorter operative time, sustained surgeon focus, and reduced risk of complications, such as meningitis [47].

Anterior pituitary insufficiency is a common preoperative condition in GPA, affecting over 50% of 57the cases in the published series. Additionally, transient or permanent hypopituitarism is a well-recognized complication of EEA. Although its incidence is comparable to that of microscopic transsphenoidal surgery, it is generally lower than that observed in transcranial procedures [48,49]. In patients with preoperative panhypopituitarism or pituitary apoplexy, postoperative hormonal recovery is often slow or may not occur at all [8,50]. Preoperative endocrine dysfunction was present in 49% of cases, with the highest incidence in the GTR group (64.7%). While many studies report endocrine recovery following tumor resection, persistent postoperative hormonal deficiencies remain a concern, particularly in patients with extensive pituitary stalk compression. In our study, new-onset hypopituitarism occurred in 10.2% of the patients, with no statistically significant differences between the groups. For functioning adenomas, remission was achieved in only 23.5% of cases in the third postoperative month. Persistent hormone hypersecretion was observed in 76.5% of the patients, highlighting the challenge of biochemical control in GPAs. The relatively low remission rate in our study is likely attributed to the high proportion of invasive tumors and the challenges associated with achieving complete tumor removal in secreting adenomas. A larger patient cohort with a longer follow-up period could provide a more comprehensive analysis of both endocrinological and surgical outcomes.

Overall, the surgical complication rate in our study was comparable to that in the previous large series on GPA surgery. The most common complications included transient diabetes insipidus (18.4%), cerebrospinal fluid (CSF) leakage (4.1%), and postoperative apoplexy (4.1%). Surgery-related mortality rates following endoscopic transsphenoidal resection of GAs range from 0% to 7.1% [51,52,53,54]. In our previous study on 205 GA cases, the mortality rate was 1.46%, with hemorrhage from the residual tumor being the primary cause of death in three patients [17]. In this series, there were no perioperative mortalities. The CSF leakage rate was minimized through a structured skull base reconstruction strategy, which included nasoseptal flaps and lumbar drainage in selected cases, a technique supported by prior literature [55,56].

### Study Limitations and Future Directions

Despite the valuable insights offered by our study, several limitations must be acknowledged. First, the relatively small sample size limited the feasibility of the multivariate analysis, restricting the application of more advanced statistical methods. Second, long-term endocrinological and functional outcomes, as well as tumor recurrence and delayed hormonal deficiencies, require extended follow-up beyond the third postoperative month for a more comprehensive assessment. To demonstrate the validity of the classification more clearly, previous studies and multicenter research are needed.

Future prospective studies incorporating advanced imaging techniques, intraoperative neurophysiological monitoring, and molecular profiling of GPAs could further refine surgical approaches and enhance patient outcomes. Moreover, the development of targeted adjuvant therapies for unresectable and aggressive pituitary tumors remains an active area of research, offering potential avenues for optimizing treatment strategies.

## 5. Conclusions

Our study highlights the critical role of combined endoscopic and transcranial approaches in the surgical management of Zone 3 GPAs, where extensive lateral and anterior cranial fossa invasion reduces the effectiveness of endoscopic endonasal surgery alone. Patients with Zone 3 tumors exhibited lower GTR rates when treated with a purely endonasal approach, highlighting the need for a multimodal strategy. The combined approach allowed for safer tumor dissection from critical neurovascular structures, optimizing resection and minimizing complications. These findings underscore the utility of the landmark-based classification system in guiding surgical decision-making and predicting resection outcomes. For this reasons, combining the EEA and transcranial approach helps overcome surgical challenges, prevent tumor apoplexy and intratumoral hemorrhage, preserve critical anatomical structures, and ultimately optimize surgical management and patient outcomes. Future studies with larger cohorts and extended follow-ups are needed to further refine patient-specific surgical strategies for complex GPAs.

## Figures and Tables

**Figure 1 cancers-17-01107-f001:**
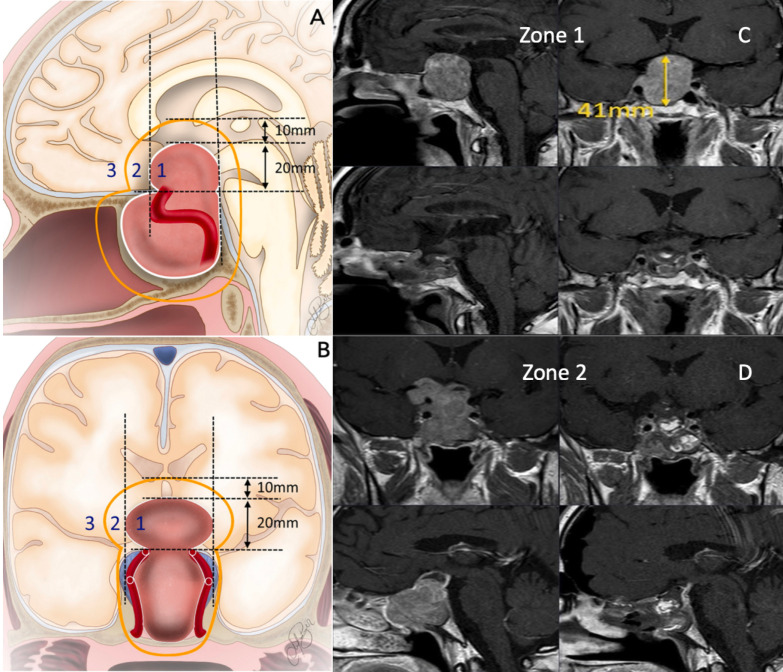
(**A**). Landmark-based classification sagittal plane view (**B**). Landmark-based classification coronal plane view (**C**). Zone 1 tumor: preoperative and postoperative NTR imaging (**D**). Zone 2 tumor: preoperative and postoperative GTR imaging.

**Figure 2 cancers-17-01107-f002:**
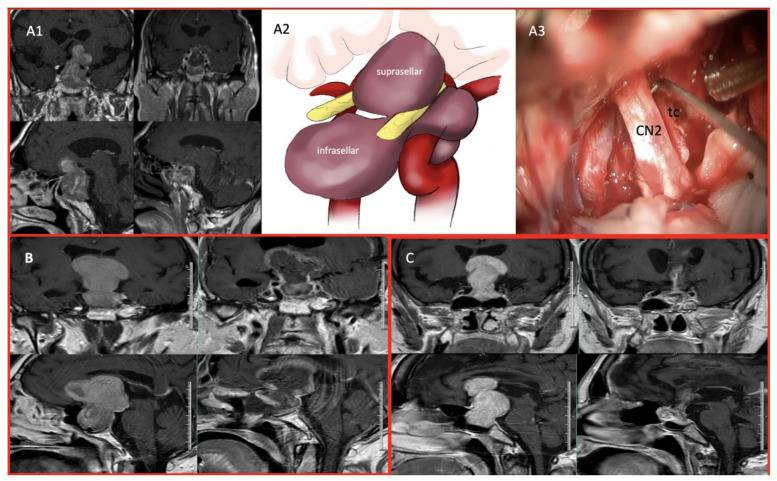
(**A1**). Preoperative MRI of a Zone 3 multilobular-shaped tumor and postoperative MRI following near-total resection using a combined approach. (**A2**). Illustrative representation of the anatomical extension way of a giant adenoma. (**A3**). Demonstration of the close relationship between optic nerve and tumor capsule during the transpterional approach. (**B**). Preoperative MRI of GA extending into the frontal lobe and lateral ventricle and postoperative MRI following NTR using a combined approach (EEA + Transpterional). (**C**). Preoperative MRI of GA extending into lateral ventricle and postoperative MRI following NTR using a combined approach (EEA + Transcallosal). (CN2: Cranial Nerve 2, tc: tumor capsule).

**Figure 3 cancers-17-01107-f003:**
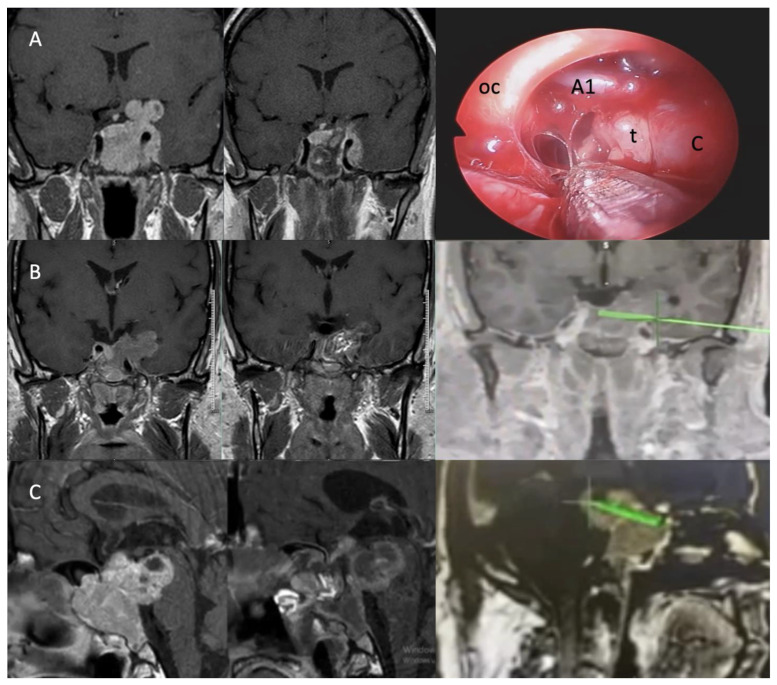
(**A**). Intraoperative view of a Zone 3 tumor with NTR and its extension posterior to the carotid artery and into the temporal region. (**B**). Demonstration of the EEA boundary in the temporal region using intraoperative navigation, along with preoperative MRI and postoperative MRI after GTR. (**C**). NTR of a retroclival-extending GA using intraoperative navigation, with preoperative and postoperative MRI images. oc: optic chiasm, t: tumor, C: carotid artery. Green Label is the border of endoscopic surgery.

**Table 1 cancers-17-01107-t001:** Clinical, radiological, and surgical characteristics of patients with giant pituitary adenomas based on the extent of resection. Categorical variables were presented with *n* (%). SD: standard deviation; IQR: interquartile range; NA: non-applicable chi-square test; TC: transcranial.

	Total (n = 49)	STR (n = 14)	NTR (n = 18)	GTR (n = 17)	*p* Value
**Age**					0.641
<40	18 (36.7)	5 (35.7)	6 (33.3)	7 (41.2)	
40–60	24 (49)	6 (42.9)	11 (61.1)	7 (41.2)	
>60	7 (14.3)	3 (21.4)	1 (5.6)	3 (17.6)	
**Ki67 (%)**					0.794
<=3	35 (71.4)	9 (64.3)	13 (72.2)	13 (76.5)	
>3	14 (28.6)	5 (35.7)	5 (27.8)	4 (23.5)	
**Gender**					0.562
Male	35 (71.4)	11 (78.6)	11 (61.1)	13 (76.5)	
Female	14 (28.6)	3 (21.4)	7 (38.9)	4 (23.5)	
**Hemianopsia**					NA
Right	10 (20.4)	3 (21.4)	2 (11.1)	5 (29.4)	
Left	5 (10.2)	2 (14.3)	2 (11.1)	1 (5.9)	
Bitemporal	34 (69.4)	9 (64.3)	14 (77.8)	11 (64.7)	
** Primary/recurrent **					NA
Primary	42 (85.7)	10 (71.4)	16 (88.9)	16 (94.1)	
Recurrent	7 (14.3)	4 (28.6)	2 (11.1)	1 (5.9)	
**P53**					0.702
Negative	31 (63.3)	9 (64.3)	10 (55.6)	12 (70.6)	
Positive	18 (36.7)	5 (35.7)	8 (44.4)	5 (29.4)	
**Proliferation**					0.300
None	27 (55.1)	7 (50)	8 (44.4)	12 (70.6)	
Increased	22 (44.9)	7 (50)	10 (55.6)	5 (29.4)	
**Adenoma Type**					0.269
Non-secretory	32 (65.3)	7 (50)	14 (77.8)	11 (64.7)	
Secretory	17 (34.7)	7 (50)	4 (22.2)	6 (35.3)	
**Secretory Type**					NA
Acth	1 (2)	0 (0)	1 (25)	0 (0)	
Gh	2 (4.1)	0 (0)	0 (0)	2 (33.3)	
Prl	13 (26.5)	6 (85.7)	3 (75)	4 (66.7)	
Tsh	1 (2)	1 (14.3)	0 (0)	0 (0)	
**Remission**					NA
No	13 (26.5)	7 (100)	4 (100)	2 (33.3)	
Yes	4 (8.2)	0 (0)	0 (0)	4 (66.7)	
**Apoplexy**					NA
No	39 (79.6)	13 (92.9)	14 (77.8)	12 (70.6)	
Yes	10 (20.4)	1 (7.1)	4 (22.2)	5 (29.4)	
**Preoperative Deficiency**					0.284
No	25 (51)	8 (57.1)	11 (61.1)	6 (35.3)	
Yes	24 (49)	6 (42.9)	7 (38.9)	11 (64.7)	
**New Postoperative Deficiency**					NA
No	44 (89.8)	11 (78.6)	17 (94.4)	16 (94.1)	
Yes	5 (10.2)	3 (21.4)	1 (5.6)	1 (5.9)	
**Max Tumor Diameter**					0.796
40–50 mm	35(71.4)	9(64.3)	13(72.2)	13((76.5)	
>50 mm	14(28.6)	5(35.7)	5(27.8)	4(23.5)	
**Surgical Approach**					NA
EEA	41 (83.7)	10 (71.4)	14 (77.8)	17 (100)	
EEA + TC	8 (16.3)	4 (28.6)	4 (22.2)	0 (0)	
**Type**					**<0.001**
None	20 (40.8)	1 (7.1)	5 (27.8)	14 (82.4)	
Complex	29 (59.2)	13 (92.9)	13 (72.2)	3 (17.6)	
**Shape**					**0.003**
Round	18 (36.7)	2 (14.3)	4 (22.2)	12 (70.6)	
Dumbbell-shaped	6 (12.2)	1 (7.1)	3 (16.7)	2 (11.8)	
Multilobular	25 (51)	11 (78.6)	11 (61.1)	3 (17.6)	
**Supra/Infra**					**0.025**
<1	31 (63.3)	6 (42.9)	10 (55.6)	15 (88.2)	
>1	18 (36.7)	8 (57.1)	8 (44.4)	2 (11.8)	
**Knosp Classification**					NA
None	2 (4.1)	1 (7.1)	0 (0)	1 (5.9)	
Knosp 1–2	16 (32.7)	0 (0)	7 (38.9)	9 (52.9)	
Knosp 3–4	31 (63.3)	13 (92.9)	11 (61.1)	7 (41.2)	
**Suprasellar Length**					**0.001**
<2 cm	30 (61.2)	4 (28.6)	10 (55.6)	16 (94.1)	
>2 cm	19 (38.8)	10 (71.4)	8 (44.4)	1 (5.9)	
**Zone**					**<0.001**
1	8 (16.3)	0 (0)	0 (0)	8 (47.1)	
2	21 (42.9)	5 (35.7)	10 (55.6)	6 (35.3)	
3	20 (40.8)	9 (64.3)	8 (44.4)	3 (17.6)	
**Tumor Extension**					NA
Sphenoid Sinus	8 (16.3)	2 (14.3)	1 (5.6)	5 (29.4)	
Chiasmatic Cistern	17 (34.7)	1 (7.1)	6 (33.3)	10 (58.8)	
Lateral Ventricle Inferior Wall	9 (18.4)	5 (35.7)	4 (22.2)	0 (0)	
Anterior Cranial Fossa	6 (12.2)	3 (21.4)	3 (16.7)	0 (0)	
Retroclival Region	5 (10.2)	2 (14.3)	2 (11.1)	1 (5.9)	
Temporal Lobe	4 (8.2)	1 (7.1)	2 (11.1)	1 (5.9)	

## Data Availability

This study was conducted in accordance with the approval provided by the relevant ethics committee.

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
