# Peer review of "Refining Endoscopic and Combined Surgical Strategies for Giant Pituitary Adenomas: A Tertiary-Center Evaluation of 49 Cases over the Past Year"

_cancers, 2025, doi:10.3390/cancers17071107_

Round 1

Reviewer 1 Report

Comments and Suggestions for Authors

Thank you for a nice description of cases in giant pituitary adenomas's surgery. The presentation is well structured, presented and illustrated. However, the conclusions seems to be too week and abstract compared to a data presented. I suggest to add some weight to the conclusions by adding recommendations of superiority of combined surgery to EEA. Especially, when this is discussed in Discussion section.

Author Response

Thank you for a nice description of cases in giant pituitary adenomas's surgery. The presentation is well structured, presented and illustrated. However, the conclusions seems to be too week and abstract compared to a data presented. I suggest to add some weight to the conclusions by adding recommendations of superiority of combined surgery to EEA. Especially, when this is discussed in Discussion section.

Answer: Thank you very much for your valuable insights. In line with your suggestions, we have made an effort to emphasize more clearly in the conclusion section that combined surgery provides significant benefits in selected cases.

  • Conclusion

For this reasons, combining the EEA and transcranial approach helps overcome surgical challenges, prevent tumor apoplexy and intratumoral hemorrhage, preserve critical anatomical structures, and ultimately optimize surgical management and patient outcomes.(Line 398-401)

Reviewer 2 Report

Comments and Suggestions for Authors

The article addresses a complex and relevant neurosurgical pathology, giant pituitary adenomas (GPAs), and suggests tailored surgical strategies based on tumor extension classification.

It highlights the advantages of the endoscopic endonasal approach (EEA) while acknowledging the need for combined or transcranial approaches in specific cases.

A retrospective analysis of 49 patients treated in a single year ensures homogeneity in surgical techniques and the surgical team.

The study employs a predefined classification system (“landmark-based classification”) to categorize tumors and correlate surgical approaches with outcomes.

It presents clear data on resection rates (GTR 34.6%, NTR 36.7%, STR 28.5%) and postoperative complications, with a detailed analysis of predictive factors for surgical success.

Appropriate statistical tests are used for group comparisons, enhancing the validity of the findings.

The discussion contextualizes the findings within the existing literature and emphasizes the importance of a multimodal strategy to optimize tumor removal and reduce morbidity.

It proposes improvements in preoperative planning through tumor classification and assessment of cavernous sinus invasion.

Areas for Improvement:

A sample of only 49 patients over one year may not be fully representative of the variability in this pathology.

The minimum follow-up period of four months is insufficient to assess tumor recurrence and the long-term progression of endocrine deficits.

The “landmark-based classification” system is introduced, but its validation in other centers or previous studies is not reported, limiting its generalizability.

Author Response

The article addresses a complex and relevant neurosurgical pathology, giant pituitary adenomas (GPAs), and suggests tailored surgical strategies based on tumor extension classification.

It highlights the advantages of the endoscopic endonasal approach (EEA) while acknowledging the need for combined or transcranial approaches in specific cases.

A retrospective analysis of 49 patients treated in a single year ensures homogeneity in surgical techniques and the surgical team.

The study employs a predefined classification system (“landmark-based classification”) to categorize tumors and correlate surgical approaches with outcomes.

It presents clear data on resection rates (GTR 34.6%, NTR 36.7%, STR 28.5%) and postoperative complications, with a detailed analysis of predictive factors for surgical success.

Appropriate statistical tests are used for group comparisons, enhancing the validity of the findings.

The discussion contextualizes the findings within the existing literature and emphasizes the importance of a multimodal strategy to optimize tumor removal and reduce morbidity.

It proposes improvements in preoperative planning through tumor classification and assessment of cavernous sinus invasion.

Areas for Improvement:

A sample of only 49 patients over one year may not be fully representative of the variability in this pathology. The minimum follow-up period of four months is insufficient to assess tumor recurrence and the long-term progression of endocrine deficits.

  • Answer: Thank you for your valuable insights on our manuscript. We had already acknowledged the limitation of having 49 patients and a short follow-up period in the limitations section. However, based on your suggestions, we have also decided to include this point in the discussion section.
  • Discussion

A larger patient cohort with a longer follow-up period could provide a more comprehensive analysis of both endocrinological and surgical outcomes.(Line 362-364)

The “landmark-based classification” system is introduced, but its validation in other centers or previous studies is not reported, limiting its generalizability.

  • Answer: Several classification systems have been proposed for giant adenomas. Recently published classification systems by Mooney et al. and Serra et al. have drawn significant attention. However, there is a need for a classification system that can indicate cases in which endoscopic resection alone may be insufficient for giant adenomas. In this study, we aimed to validate the classification we previously proposed. However, acknowledging the need for multicenter studies and larger case series for broader acceptance of the classification, this point has been included in the limitations section.
  • Study Limitations and Future Directions

To demonstrate the validity of the classification more clearly, previous studies and multicenter research are needed.(Line 380-381)

Reviewer 3 Report

Comments and Suggestions for Authors

Very impressive series of pituitary tumor in one year.

1. The statistical analysis of extent of tumor resection should be calculated differently.  For example, for zone 1 tumor, GTR is 100% ( 8 of 8). For zone 2 tumor, GTR is 28.6% (6 of 21 in zone 2), NTR 47.6% (10 of 21) and STR 23.8 % (5 of 21).  For zone 3 , GTR is 15 % ( 3 of 20 in zone 3) , NTR 40 % (8 of 20) and STR is 45 % (9 of 20).  This clearly demonstrated that GTR is 100 % for zone 1 tumor whereas rate of complete resection dropped with zone 2-3 tumor (28.6 % for zone 2 , 15 % for zone 3) and rate of incomplete resection ( NTR and STR ) increases substantially with higher grade tumor invasiveness and extension, 71.4 % for zone 2 and 85% for zone 3.

2. For Knosp 0, there were 2 cases but only 1 had GTR, there other was STR.  What other characteristic of that Knosp 0 case that precluded it form GTR ?

Tumor type Total STR NTR GTR

Max Tumor diameter        
40-50 mm 35 9 (25.7) 13 (37.1) 13 (37.1)
>50 mm 14 5 (35.7) 5 (35.7) 4 (28.6)

Type        
None 20 1 (5.0) 5 (25.0) 14 (70.0)
Complex 29 13 (44.8) 13 (44.8) 3 ( 10.3)

Shape        
Round 18 2 (11.1) 4 (22.2) 12 (66.7)
Dumbell 6 1 (16.0) 3 (50.0) 2 (33.3)
Multilobar 25 11 (44.0) 11 (44.0) 3 (12.0)

Knosp        
None 2 (4.0) 1 (50.0) 0 1 (50.0)
Knosp 1-2 16 (32.7) 0 7 (43.7) 9 (56.3)
Knosp 3-4 31 (63.3) 13 (41.9) 11 (35.5) 7 (22.6)

Suparsellar length        

<2 cm 30      
>2cm 19      

Zone        
1 8 (16.3) 0 0 8 (100.0)
2 21 (42.9) 5 (23.8) 10 (47.6) 6 (28.6)
3 20 (40.8) 9 (45.0) 8 (40.0) 3 (15.0)

3. In Table 1, for max tumor diameter, there should be 14 of >50 mm, not 15.

Author Response

Very impressive series of pituitary tumor in one year.

-Answer: Thank you for your kind and thoughtful comments on our manuscript. Based on your critiques, we have made additions that we believe will add value to our work.

  1. The statistical analysis of extent of tumor resection should be calculated differently.  For example, for zone 1 tumor, GTR is 100% ( 8 of 8). For zone 2 tumor, GTR is 28.6% (6 of 21 in zone 2), NTR 47.6% (10 of 21) and STR 23.8 % (5 of 21).  For zone 3 , GTR is 15 % ( 3 of 20 in zone 3) , NTR 40 % (8 of 20) and STR is 45 % (9 of 20).  This clearly demonstrated that GTR is 100 % for zone 1 tumor whereas rate of complete resection dropped with zone 2-3 tumor (28.6 % for zone 2 , 15 % for zone 3) and rate of incomplete resection ( NTR and STR ) increases substantially with higher grade tumor invasiveness and extension, 71.4 % for zone 2 and 85% for zone 3.

Answer: To maintain the coherence of the data, we did not make any changes to the table. However, we have incorporated your suggested example into the results section.

Results

For zone 1 tumors, GTR is 100% (8/8). In zone 2, GTR is 28.6% (6/21), NTR 47.6% (10/21), and STR 23.8% (5/21). In zone 3, GTR drops to 15% (3/20), while NTR and STR rise to 40% (8/20) and 45% (9/20), respectively. These findings indicate a decline in complete resection from zone 1 to zone 3, with incomplete resection rates increasing as tumor invasiveness and extension progress (71.4% in zone 2, 85% in zone 3). (Line 229-233)

  1. For Knosp 0, there were 2 cases but only 1 had GTR, there other was STR.  What other characteristic of that Knosp 0 case that precluded it form GTR ?

Answer: The Knosp 0 case that underwent subtotal resection was an isolated suprasellar multilobulated adenoma with superior extension. This was the reason for the lower resection rate. This point has been added to the results section.

Results

Among two patients with Knosp 0 adenomas, one underwent GTR, while the other underwent STR. The patient who underwent STR had an isolated suprasellar multilobulated adenoma. (Line 223-225)

  1. In Table 1, for max tumor diameter, there should be 14 of >50 mm, not 15.

Answer: It has been corrected in accordance with your suggestion.

Round 2

Reviewer 3 Report

Comments and Suggestions for Authors

Acceptable revision.